**Data Availability Statement:** All relevant data are within the paper and its Supporting Information files.

**Funding:** The author(s) received no specific funding for this study.

# Survey from 61,748 schools in four States of India on sale of tobacco products near schools

**Ashima Sarin**📷[1]☯*, **Sanjay Seth**[1]☯, **Barkha Sethi**[1]☯, **Hitesh R. Singhavi**[2]

**1** Sambandh Health Foundation, Gurugram, India, **2** Fortis Hospital, Mulund, Mumbai, India

☯ These authors contributed equally to this work.

* ashima.sarin@sambandhhealth.org

## Abstract

### Background

Children form the most vulnerable strata of the society and the tobacco industry is known to target them. Article 16 of the Framework Convention of Tobacco Control (FCTC) calls for prohibition of tobacco sales to and by minors. Although interventions to stop such sales are based on sound science, it is widely acknowledged that many countries find implementation, full of challenges. In India, sales near educational institutions are banned by law, Section 6b of the Cigarettes & Other Tobacco Products Act (COTPA). We conducted a survey of violations in four states [Andhra Pradesh (AP), Karnataka (KA), Meghalaya (ML), Uttar Pradesh (UP)] of India to report the number of violations and to assess if there was an association between the schools with violations and variables such as gender, size, category, location of schools.

### Methods

Schools in these States were asked to report the number of shops selling tobacco within 100 yards on an App circulated to all schools (289,392 in number). Chi-square tests, univariate and multivariate logistic regression performed to find association between schools with violations by Category of School, Size of School, Gender of students and location (Urban/Rural).

### Findings

Responses were received from 61,748 (21.3%) schools of which 16,193 (26.2%) reported violations. It was observed that the percentages of the schools with violations were similar to the prevalence of tobacco usage in the state. Four states AP, KA, ML, UP reported violations 22.2%, 17.5%, 42.9% and 31.4% respectively. On chi-square tests, there was a significant association for the states of KA and ML with regards to variables like size, category, location of schools (*p-value <0.001*). For AP, all variables were significantly different (*p-value <0.001*) while for UP variables like size and gender were significant. On logistic regression, there was significant association between the variables like size (>100),

**Competing interests:** The authors have declared that no competing interests exist.

category (Upper Primary) and location (urban) for tobacco shops violations in both KA and ML except for the school category that was secondary in ML. While for AP and UP, only size (>100), location (urban) of schools have a significant association with the violations of tobacco shops. Logistic regression of pooled data of four states school size (>100) and school category (primary) and location (urban) had significantly higher association of violation of tobacco shops.

## Interpretation

This is the first large survey with responses from almost all parts of the four states in India. This study shows significant association with the size, category and location of schools. We anticipate that the lists of schools which have such violations can be used by enforcement agencies to take focused action. Such models will help develop effective tobacco control policies in developing countries with large populations where implementation remains a big challenge.

## Introduction

India has a tobacco control law known as the Cigarettes & Other Tobacco Products Act (COTPA). Section 6(b) of COTPA states that tobacco cannot be sold within 100 yards of any educational institution. The aim of the Act is to reduce the prevalence of tobacco use in the country and to protect the youth. According to the Global Adult Tobacco Survey (GATS 2016), there were more than 266.8 million (28.6%) tobacco users (aged 15+) and the Ministry of Health and Family Welfare (MoHFW) estimates that 1.35 million deaths every year are attributable to tobacco use in India [1] As per the Global Youth [2] Tobacco Survey (GYTS 2019) by the MoHFW, 8.5% children aged 13–15 years consume tobacco in some form in India [3].

Children form the most vulnerable strata of the society and the tobacco industry is known to target them [4].

This industry invests heavily in advertisements and strategies influencing youth to make them believe that the usage of tobacco is a lifestyle choice. Tobacco sale near schools provides easy access to children [5]. The Section 6(b) of the Indian Tobacco Control Law, i.e., the Cigarettes and Other Tobacco Products Act (COTPA) [6], prohibits the sales of tobacco within 100 yards of the school premises. However, enforcement of the COTPA Section 6(b), remains a huge challenge despite of various Indian High Courts [7–12] have given orders over the years to stop sales of tobacco near schools.

A study done in a metropolitan city of India (Bengaluru) on the compliance of Section 6b of COTPA [13] showed that 76.2% of the educational institutions had shops near educational institutions. Another study by Irfan et al in 2020, the data was collected from 70 educational institutions in capital of India (Delhi). It was observed that the compliance to the Section 6b was 52.85% [14]. In an earlier study, we had done 'dip-stick' surveys [15] to determine the occurrence of shops selling tobacco near schools, however, a large-scale survey truly representing the entire state was awaited.

Through this survey in four states of India, i.e., Andhra Pradesh, Karnataka, Meghalaya and Uttar Pradesh, our primary goal was to find out the percentage of schools where COTPA section 6(b) is being violated. Secondly, also to find out whether there was an association between

the schools with shops selling tobacco within 100 yards and variables such as size, category, gender, location of schools. Finally, we also wanted to observe whether there is any relationship between the overall prevalence of tobacco use in the state and percentage of schools with Section 6(b) violation.

## Materials and methods

The survey was done on-line by us using our web-based App, wherein a principal/teacher representing the school was asked to report the number of shops selling tobacco products within 100 yards of their schools.

The data collection in these states was a component of the ToFEI (Tobacco Free Educational Institution) program being rolled out there by us. The Tobacco Free Educational Institutions is a Ministry of Health & Family Welfare, Government of India, program of which Revised Guidelines were issued in May 2019. This program requires every educational institution in the country to do a number of anti-tobacco activities, one of which is for schools to report if there are any shops selling tobacco within 100 yards. In the four target states, Orders have been issued by the State Education Departments to implement the ToFEI program and hence, report the COTPA Section 6b violations on our App by the Principals and Teachers in schools. As the study did not involve research on human subjects, no ethics approval or consent was required.

The initial schools list with gender, category, size and whether urban/rural was taken from the Unified District Information System for Education (UDISE) 2017–18 [16]. UDISE is a Central Government database available in public domain. Since the UDISE database was from 2017–18, for this study, the initial schools list from the UDISE [16] portal was updated with the latest school lists collected from Education Departments in the four States of India. The states are Andhra Pradesh, Karnataka, Meghalaya and Uttar Pradesh.

The App was distributed to all schools in the target states through the respective Education Departments. On the App, the schools, with their UDISE number, reported the number of shops selling tobacco near their schools (0, 1, 2 or 3+). The UDISE database provided information of the same schools about gender, category, size and whether urban/rural. The schools covered in this study from the four states totalled 2,89,392. Of these 61,748 schools, i.e., our sample, reported on COTPA Section 6b violations, that is, 21.3%.

Definitions: Gender indicated whether the school was Girls, Boys or Co-educational. As to "Category" the highest school level was taken. For instance, if a school has a 'Primary' and also 'Upper Primary' section, it has been counted in 'Upper Primary' category. "Size" was classified as: (I) Less than 30 students, (II): 30–100 students and (III): more than 100 students. Urban/ Rural: In UDISE, "Urban" is defined as within Municipalities (a town or district that has local government) and Rural as outside Municipalities. Data has been collected from 1st June 2020 to 30th October 2021. From the databases, the compilation was done using Google Sheets and dashboards were created to generate reports for this study. In addition to the UDISE portal, data from National Health Family Survey (NHFS) [17], GATS, GYTS was used for comparison.

The data was analysed using SPSS version 25.0 (IBM SPSS Statistics, Version 25.0, Armonk, NY: IBM Corp.) for Categorical variables, viz. size, school category, gender and location were given as percentages and their association with presence of shop selling tobacco within 100 yards tested using chi-square test for association. The univariate and multivariate logistic regression analyses was performed to ascertain the effects of size, school category, gender and location on the likelihood on outcome variable, i.e., "presence of shop selling tobacco within 100 yards" by estimating odds ratios and their 95% confidence intervals. For the multivariate

**Table 1. Geographical areas covered & schools reported statewise.**

| State | Districts | | Blocks | | | Schools | | |
|---|---|---|---|---|---|---|---|---|
| | Total | Surveyed | Total | Reported | %age | Total | Reported on COTPA 6B | %age |
| Andhra Pradesh | 13 | 13 | 659 | 648 | 98.3% | 61,954 | 18,445 | 29.8% |
| Karnataka | 34 | 34 | 204 | 203 | 99.5% | 72,527 | 15,737 | 21.7% |
| Meghalaya | 11 | 11 | 48 | 48 | 100.0% | 14,744 | 5,950 | 40.4% |
| Uttar Pradesh | 75 | 75 | 917 | 864 | 94.2% | 1,40,167 | 21,616 | 15.4% |
| TOTAL | **133** | **133** | **1,828** | **1,763** | **96.4%** | **2,89,392** | **61,748** | **21.3%** |

Note: Blocks are sub-district areas as defined by Education Department.

analysis, statistically significant variables in the univariate analysis were included in multivariate logistic regression model. The p-value <0.05 was considered as statistically significant.

## Results

The schools covered in the four States totalled 2,89,392. Of these 61,748 schools reported on COTPA Section 6b violations, that is, 21.3%. State-wise it varied from 15.4% in Uttar Pradesh to 40.4% in Meghalaya. As Table 1 shows schools from every district and 96.4% of Blocks reported, indicating that the survey covered almost the entire geographical areas of these States. In India, the states are divided, for administration purposes, into Districts and sub-districts which are known by different names in the country such as Taluk, Block, Tehsil, Mandal etc [18]. Since routine enforcement activities, like policing, is done at block level, it is assumed that compliance with laws such as Section 6b will be fairly uniform across a block and hence, if schools from most blocks have reported, the data is fairly representative.

The Principal/Teacher reported the number of shops selling tobacco within 100 yards on the App from 61,748 schools, of which 16,193 (26.2%) schools reported that there was one or more shops (called Tobacco-shops herein) selling tobacco products. Schools from all 133 districts and 1,763 Blocks (out of 1868) of the four States reported activity on the App. All 133 districts and 88.5% of the blocks had Tobacco-shops. Table 1.

A comparison of the schools with Tobacco-shops with the GATS 2016–17 [19], GYTS 2019 [20] and NFHS-5 (2019–21) was done Table 2.

In the states of Andhra Pradesh and Karnataka, the percentage of schools with Section 6b violations were 22.2% and 17.5% respectively, whereas the prevalence of tobacco use, as per GATS 2017, in these States was 20.0% and 22.8% in 2016–17. In Meghalaya and Uttar Pradesh, schools with Section 6b violations are 42.9% and 31.4% respectively, whereas tobacco use prevalence percentages are 47% and 35.5%.

**Table 2. Schools with shops selling tobacco versus tobacco prevalence.**

| Sn | Description | Andhra Pradesh | Karna-taka | Megh-alaya | Uttar Pradesh |
|---|---|---|---|---|---|
| 1 | Schools with shops selling Tobacco within 100 yards | 22.2% | 17.5% | 42.9% | 31.4% |
| 2 | GATS 2017—current users | 20.0% | 22.8% | 47.0% | 35.5% |
| 3 | NFHS-5 (2019–20) | 13.2% | 17.7% | 42.8% | 25.9% |
| 4 | GYTS-4—current users | 2.6% | 1.2% | 34.0% | 23.0% |

Notes

a GATS 2017 = Global Adult Tobacco Survey, people aged 15+. Data collected in 2016 & published in 2017.

b NFHS-5 (2019–20) = The National Family Health Survey. People aged 15+. Fieldwork was done in 2019 and published in 2020.

c GYTS-4 = The Global Youth Tobacco Survey. People aged 13–15. Fieldwork done in 2019 and published 2021.

**Table 3. Chi-square tests for association (Karnataka & Meghalaya).**

| Variable | Karnataka | | | Meghalaya | | |
|---|---|---|---|---|---|---|
| | With T-shops | N = 15,615 | | With T-shops | N = 5,869 | P-Value* |
| *Size (students per school)* | | | | | | |
| <30 | 580 (21.2%) | 4,262 (27.3%) | < 0.001 | 683 (27.1%) | 1,846 (31.5%) | < 0.001 |
| 30–100 | 951 (34.8%) | 5,303 (34.0%) | | 1,394 (55.2%) | 3,188 (54.3%) | |
| >100+ | 1,205 (44.0%) | 6,050 (38.7%) | | 446 (17.7%) | 835 (14.2%) | |
| *School Category (highest class in school)* | | | | | | |
| Primary only | 851 (31.1%) | 5,762 (36.9%) | < 0.001 | 1,469 (57.5%) | 3,633 (61.1%) | < 0.001 |
| Upper Primary | 1,343 (49.1%) | 6,616 (42.4%) | | 730 (28.6%) | 1,642 (27.6%) | |
| Secondary | 497 (18.2%) | 2,968 (19.0%) | | 288 (11.3%) | 544 (9.1%) | |
| Higher Secondary | 45 (1.6%) | 269 (1.7%) | | 67 (2.6%) | 131 (2.2%) | |
| *Location* | | | | | | |
| Rural | 2,128 (77.8%) | 12,881 (82.5%) | < 0.001 | 2,267 (88.8%) | 5,487 (92.2%) | < 0.001 |
| Urban | 608 (22.2%) | 2,734 (17.5%) | | 287 (11.2%) | 463 (7.8%) | |
| *Gender* | | | | | | |
| Co-educational | 2,649 (96.8%) | 15,207 (97.4%) | 0.115 | 2,504 (99.3%) | 5,837 (99.5%) | 0.165 |
| Girls | 61 (2.2%) | 280 (1.8%) | | 18 (0.7%) | 30 (0.5%) | |
| Boys | 26 (1.0%) | 128 (0.8%) | | 1 (0.0%) | 2 (0.0%) | |
| **Remarks** | 15,737 respondents less 122 with mistakes not considered in analysis | | | 5,950 respondents less 81 with mistakes not considered in analysis | | |

(Table 3).

By School Category: On chi-square test of association, in the state of Karnataka, there was significant difference in reported T-shops according school category as depicted in Table 3. On logistic regression modelling, upper primary school category had significant higher T-shops as compared to primary only (OR: 1.15) while lower number shops were present in only primary schools as compared to secondary (OR.0.835) and higher secondary (0.829) (Table 5). On chi-square test of association, in the state of Meghalaya, there was significant difference in reported T-shops according school category as depicted in Table 3. On logistic regression modelling, primary school category had significant higher T-shops as compared to upper primary (OR: 1.177) and secondary (1.519) while T-shops were similar as compared to higher secondary (OR-1.032) (Table 5). On chi-square test of association, in the state of Uttar Pradesh, there was no significant difference in reported T-shops according school category as depicted in Table 4. On chi-square test of association, in the state of Andhra Pradesh, there was significant difference in reported T-shops according school category as depicted in Table 4. On pooled analysis for all four states, chi-square test of association, there was significant difference in reported T-shops according school category as depicted in Table 7.

By the size of School: Three categories were studied: Schools with (I) Less than 30 students, II: 30–100 students and III: more than 100 students. On chi-square test of association, in the state of Karnataka, there was significant difference in reported T-shops according size of the school as depicted in Table 3. On logistic regression modelling, size of the school greater than 100 had odds ratio of 1.466 while 30–100 had odds ratio of 1.305 as compared to <30 (Table 5). On chi-square test of association, in the state of Meghalaya, there was significant difference in reported T-shops according size of the school as depicted in Table 3. On logistic regression modelling, size of the school greater than 100 had odds ratio of 1.709 while 31–100 had odds ratio of 1.323 as compared to <30 (Table 5). On chi-square test of association, in the state of Uttar Pradesh, there was significant difference in reported T-shops according size of

**Table 4. Chi-square tests for association (Andhra Pradesh & Uttar Pradesh).**

| Variable | Andhra Pradesh | | | Uttar Pradesh | | |
|---|---|---|---|---|---|---|
| | With T-shops | N = 18,186 | P-Value* | With T-shops | N = 21,600 | P-Value* |
| *Size (students per school)* | | | | | | |
| <30 | 1,170 (28.9%) | 6,098 (33.5%) | < 0.0001 | 457 (6.7%) | 1,824 (8.4%) | < 0.001 |
| 30–100 | 1,643 (40.6%) | 7,044 (38.7%) | | 3,449 (50.9%) | 11,569 (53.6%) | |
| >100+ | 1,233 (30.5%) | 5,044 (27.8%) | | 2,877 (42.4%) | 8,207 (38.0%) | |
| *School Category (highest class in school)* | | | | | | |
| Primary only | 2,696 (66.6%) | 12,473 (68.6%) | 0.012 | 4,678 (69.0%) | 14,734 (68.2%) | 0.263 |
| Upper Primary | 500 (12.4%) | 2,131 (11.7%) | | 2,103 (31.0%) | 6,861 (31.8%) | |
| Secondary | 816 (20.2%) | 3,463 (19.0%) | | 4 (0.0%) | 14 (0.1%) | |
| Higher Secondary | 34 (0.8%) | 119 (0.7%) | | | | |
| *Location* | | | | | | |
| Rural | 3,525 (87.1%) | 16,335 (89.8%) | < 0.0001 | 6,503 (95.8%) | 21,030 (97.3%) | 0.612 |
| Urban | 521 (12.9%) | 1,851 (10.2%) | | 282 (4.2%) | 579 (2.7%) | |
| *Gender* | | | | | | |
| Co-educational | 3,925 (97.0%) | 17,813 (98.0%) | <0.0001 | 6,712 (98.9%) | 21,464 (99.3%) | < 0.001 |
| Girls | 85 (2.1%) | 267 (1.4%) | | 46 (0.7%) | 88 (0.4%) | |
| Boys | 36 (0.9%) | 106 (0.6%) | | 27 (0.4%) | 57 (0.3%) | |
| **Remarks** | 18,445 respondents less 258 with mistakes not considered in analysis | | | 21,616 respondents less 16 with mistakes not considered in analysis | | |

the school as depicted in Table 4. On logistic regression modelling, school category greater than 100 had odds ratio of 1.644 while 31–100 had odds ratio of 1.288 as compared to <30 (Table 6). On chi-square test of association, in the state of Andhra Pradesh, there was significant difference in reported T-shops according size of the school as depicted in Table 6. On logistic regression modelling, size of the school greater than 100 had odds ratio of 1.261 while 31–100 had odds ratio of 1.265 as compared to <30 (Table 6). On pooled analysis, chi-square test of association, there was significant difference in reported T-shops according size of the school as depicted in Table 7. On logistic regression modelling, size of the school greater than 100 had odds ratio of 1.501 while 31–100 had odds ratio of 1.639 as compared to <30 (Table 8).

**Table 5. Logistic regression (Karnataka & Meghalaya).**

| Variable | Karnataka | | Meghalaya | |
|---|---|---|---|---|
| | Odds ratio [Upper—Lower Bound | P-value | Odds ratio [Upper—Lower Bound | P-value |
| *Size (students per school)* | | | | |
| <30 | Ref | | Ref | |
| 30–100 | 1.305 [1.144 to 1.489] | < 0.001 | 1.323 [1.175 to 1.489] | < 0.001 |
| >100+ | 1.466 [1.259 to 1.707] | < 0.001 | 1.709 [1.436 to 2.035] | < 0.001 |
| *School Category (highest class in school)* | | | | |
| Primary only | Ref | | Ref | |
| Upper Primary | 1.155 [1.02 to 1.308] | 0.023 | 1.177 [1.044 to 1.326] | 0.008 |
| Secondary | 0.835 [0.715 to 0.976] | 0.023 | 1.519 [1.257 to 1.836] | <0.001 |
| Higher Secondary | 0.829 [0.586 to 1.173] | 0.289 | 1.032 [0.704 to 1.512] | 0.872 |
| *Location* | | | | |
| Rural | Ref | | Ref | |
| Urban | 1.375 [1.238 to 1.528] | < 0.001 | 2.009 [1.642 to 2.458] | < 0.001 |

**Table 6. Logistic regression (Andhra Pradesh & Uttar Pradesh).**

| Variable | Andhra Pradesh | | Uttar Pradesh | |
|---|---|---|---|---|
| | Odds ratio [Upper—Lower Bound | P-value | Odds ratio [Upper—Lower Bound | P-value |
| *Size (students per school)* | | | | |
| <30 | Ref | | Ref | |
| 30–100 | 1.261 [1.159 to 1.373] | < 0.001 | 1.288 [1.149 to 1.442] | < 0.001 |
| >100+ | 1.265 [1.151 to 1.390] | < 0.001 | 1.644 [1.464 to 1.845] | < 0.001 |
| *Gender* | | | | |
| Co-educational | Ref | | Ref | |
| Girls | 1.435 [1.1 to 1.872] | 0.008 | 2.293 [1.493 to 3.522] | <0.001 |
| Boys | 1.624 [1.081 to 2.441] | 0.02 | 1.950 [1.154 to 3.294] | 0.013 |
| *Location* | | | | |
| Rural | Ref | | Ref | |
| Urban | 1.328 [1.189 to 1.484] | < 0.001 | 2.147 [1.817 to 2.536] | < 0.001 |

By Gender: Schools were also classified as boys only, girls only and co-educational ones. In Meghalaya, Karnataka, on chi-square test of association, there was no significant difference in reported T-shops according the gender based schools as depicted in Table 3. The girls and boys schools had similar percentages of such violations. However, in Uttar Pradesh and Andhra Pradesh, there was significant difference between the availability of tobacco shops in co-educational, girls only and boys only school. On logistics regression analysis, in Andhra Pradesh, boys-only school (OR -1.624), girls-only (OR-1.435) had significantly higher reported T-shops as compared to co-educational Table 6. While for Uttar Pradesh, logistics regression analysis analysed to have higher T-shops for girls only school (OR- 2.29), boys only school (OR- 1.95) as compared to co-ed. (Table 6). On pooled analysis of all the four states, Girls only schools had higher tobacco shops with OR of 1.229 and boys only being 1.218 as compared to co-ed (Table 8).

**Table 7. Chi-square tests for association (four states—pooled data).**

| Variable | Pooled Report | | |
|---|---|---|---|
| | With T-shops | N = 61270 | P-Value* |
| **Remarks** | 61,748 respondents less 478 with mistakes not considered in analysis | | |
| **Size (students per school)** | | | |
| <30 | 2,890 (18.0%) | 14,030 (22.9%) | < 0.001 |
| 30–100 | 7,437 (46.2%) | 27,104 (44.2%) | |
| >100+ | 5,761 (35.8%) | 20,136 (32.9%) | |
| **School Category (highest class in school)** | | | |
| Primary only | 9,694 (60.1%) | 36,602 (59.7%) | <0.001 |
| Upper Primary | 5,445 (33.8%) | 20,599 (33.6%) | |
| Secondary | 858 (5.3%) | 3,683 (6.0%) | |
| Higher Secondary | 124 (0.8%) | 476 (0.8%) | |
| **Location** | | | |
| Rural | 14,423 (89.5%) | 55,733 (90.8%) | < 0.001 |
| Urban | 1,698 (10.5%) | 5,627 (9.2%) | |
| **Gender** | | | |
| Co-educational | 15,790 (98.1%) | 60,321 (98.4%) | 0.002 |
| Girls | 210 (1.3%) | 665 (1.1%) | |
| Boys | 90 (0.6%) | 293 (0.5%) | |

**Table 8. Logistic regression (four states: Pooled data).**

| Variable | Pooled Data | |
|---|---|---|
| | Odds ratio [Upper—Lower Bound] | P-value |
| **Size (students per school)** | | |
| <30 | Ref | |
| 30–100 | 1.501 [1.1428 to 1.577] | < 0.001 |
| >100+ | 1.639 [1.552 to 1.731] | < 0.001 |
| **School Category (highest class in school)** | | |
| Primary only | Ref | |
| Upper Primary | 0.923 [0.885 to 0.962] | < 0.001 |
| Secondary | 0.651 [0.609 to 0.695] | < 0.001 |
| Higher Secondary | 0.810 [0.665 to 0.986] | 0.036 |
| **Gender** | | |
| Co-educational | Ref | |
| Girls | 1.229 [1.039 to 1.455] | 0.016 |
| Boys | 1.218 [0.948 to 1.566] | 0.123 |
| **Location** | | |
| Rural | Ref | |
| Urban | 1.233 [1.159 to 1.312] | < 0.001 |

By location: On chi-square test of association, there was significant difference in reported T-shops according location of the school for all the four states as depicted in Table 3. On logistic regression modelling, location of the school (urban) had higher tobacco shops in Karnataka (OR:1.375), Meghalaya (OR: 2.009), Uttar Pradesh (OR- 2.147) and Andhra Pradesh (OR:1.328) [Tables 3 and 4]. On pooled analysis, urban areas had significantly higher T-shops with odds ratio of 1.233 (Table 8).

## Discussion

This large survey of COTPA Section 6b violations in four states of India shows that sale of tobacco products near schools is high and remains a matter of concern for tobacco control. More than a quarter of the respondents reported violation of Section 6b of COTPA. The study also indicates that higher the prevalence of tobacco users in the state, higher the percentage of schools Section 6b violations.

It is believed that tobacco industry strategizes on infusing tobacco practices in the formative years of life leading to lifelong addiction. One of the actions to target youth is to provide greater and easy access to tobacco by ensuring that there is more sale of tobacco near educational institutions. It may well be said that such sale is only a consequence of demand-supply dynamics and not a tobacco industry stratagem. In either case, Article 16 of the Framework Convention of Tobacco Control [21] calls for prohibition of tobacco sales to and by minors. Although interventions to stop such sales are based on sound science, it is widely acknowledged that many countries find implementation, full of challenges. In India, sales near educational institutions are banned by law, Section 6b of COTPA. There are also severe penalties under Section 77 of the Juvenile Justice Act 2016 [22] for any sales to minors. Further, Ministry of Health & Family Welfare, Government of India has made stopping such sales as a component of the Guidelines for Tobacco Free Educational Institutions (ToFEI) program, which seeks to protect youth from tobacco and has funded State Governments for more than a decade to implement this program under the National Tobacco Control Program (NTCP). However, in spite of the laws and programs, there is a concern about sales near schools

amongst tobacco control activists as evidenced by the repeated Public Interest Litigation (PIL) filed in High Courts across India over the years. Dip stick surveys [15] have also indicated noncompliance.

Hence, monitoring the sales near schools is likely an essential step in effective implementation of the tobacco control laws. Surveys like this provide insights for design of effective and sustainable programs for stopping sales near educational institutions. In our previous study [15], we had done a "dip-stick" survey of 307 schools, 69% of which had violations of section 6b. In the previous study [15], data was collected by a tobacco control coordinator and schools within a 10 km radius of the city-centre were chosen. In another study by Goel et al [23] found that 32.5% had tobacco shops within the radius of 100 yards. The current study is not only vastly larger in scope but provides a model for on-going surveillance and also provides actionable inputs to enforcement agencies to stop such sales. This data collection was done as part of ToFEI implementation roll out, however, similar data can be collected by the Education Departments using freeware like Google Forms.

This study indicates that higher the prevalence of tobacco users in the state, higher the percentage of schools with tobacco shops (Section 6b violations). While a cross-sectional study like this may not allow comments on the causal relationship between the density of shops around schools and prevalence, this is a clear indicator of the tobacco control efforts in the States. With 26.2% shops near educational institutions in the four states shows lack of enforcement of COTPA Section 6.

According to the Global Youth Tobacco Survey [2], the median age of initiation of consumption of smokeless tobacco products was 9.9 years. GYTS describes this age group as a global standard system for monitoring youth tobacco use. Therefore, we assessed whether the tobacco shop violations were influenced by the "category" of schools. We found primary schools to be having higher number of tobacco shop violations. Our study shows that there is a paradigm shift in the focus of tobacco industry towards younger population groups.

The tobacco industry does focus on school students, however, there is no evidence in the literature to suggest that. Our study shows that there was significant association between the size of the schools and tobacco shop violations. There was a significant higher number of tobacco shops around the schools with larger number of students. This suggests that tobacco industry concentrates on school students.

A study done by McCarthy et al [24] analysed density of tobacco retailers around schools and its effect of tobacco use amongst its students found experimental smoking higher in urban areas in California, USA. However, there is no Indian literature available to back the same strategy that tobacco industry adopts to lure school students in urban areas. Our study shows that urban schools had significantly higher number of tobacco shops in all four states of India. Thus, concluding the same tactic of tobacco industry to lure the urban school students as is the case in the above international study.

Another finding was that in all four states, schools with Section 6b violations near girls' schools were considerably higher than those around the co-educational schools, or even the boys schools in three States, i.e., Karnataka, Meghalaya and Uttar Pradesh. In Uttar Pradesh, this is already reflecting in the GYTS-4 data which shows that 24% of girls are using tobacco as against 22% boys. However, these findings become less relevant considering the difference in the number of 'girls only' (1.3%), 'boys only' (0.6%) as compared to co-ed schools (98.1%). It paves way for more research in this direction.

Our survey is the first study to get responses from a large sample from every nook and corner of these four States. The density of shops selling tobacco clearly indicates that violations of Section 6b of COTPA remains a huge challenge.

## Limitations

The distance of tobacco shops from the school boundary was not actually measured and is subject to the accuracy of the judgement of those principals and teachers. Once the results of this online survey are provided to the enforcement authorities and they start taking action, the reporting accuracy will increase. Data is subject to recall bias and data-entry errors as data is self-reported. However, app-based collection of data makes it easy and real-time. Study has been in four states via App, thus, data is not representative of the country yet it represents those four states. While a small pilot was done in Meghalaya to check the results of the on-line survey against on-ground situation, a statistical validation of the App remains to be done.

## Recommendations

The study suggests that app-based collecting data on shops selling tobacco nearby, directly from schools, is a good method to monitor the level of violations of Section 6b of COTPA. This method will periodically provide lists of schools which have such violations and these can be used by enforcement agencies to take focused action. This may well provide a sustainable and reliable system for protecting children from access to tobacco. Policymakers can take a note of the variables such as size of the school, category of the schools and location of schools while making policies for tobacco control.

## Conclusion

Our study showcases the rampant violation of Section 6b in four states of India. Overall violation was 26.2% in these states. The number of tobacco shops were influenced by the location, category and size of schools. Our study also suggests that it corresponds to the incidence of overall tobacco use in those states.

## Supporting information

**S1 Raw data.**
(XLSX)

**S2 Raw data.**
(XLSX)

## Acknowledgments

The authors would like to thank Ministry of Health & Family Welfare, Government of India; Department of Health, Government of Meghalaya; Department of Education, Governments of Andhra Pradesh, Karnataka, Meghalaya, Uttar Pradesh; Narayana Cancer Care, Bangalore, Karnataka; Anil Patil and Akshay Patil. The conclusions and recommendations are those of authors and do not represent views of any organization.

## Author Contributions

**Conceptualization:** Ashima Sarin, Sanjay Seth.

**Data curation:** Barkha Sethi, Hitesh R. Singhavi.

**Formal analysis:** Hitesh R. Singhavi.

**Methodology:** Ashima Sarin, Sanjay Seth, Hitesh R. Singhavi.

**Project administration:** Sanjay Seth, Hitesh R. Singhavi.

**Software:** Barkha Sethi.

**Supervision:** Ashima Sarin, Sanjay Seth.

**Writing – original draft:** Ashima Sarin.

**Writing – review & editing:** Ashima Sarin, Sanjay Seth, Hitesh R. Singhavi.

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
