## [Decision Letter · Decision Letter 0]

29 Mar 2023

PONE-D-22-32913SURVEY FROM 61,748 SCHOOLS IN FOUR STATES OF INDIA ON SALE OF TOBACCO PRODUCTS NEAR SCHOOLSPLOS ONE

Dear Dr. Sarin,

Thank you for submitting your manuscript to PLOS ONE. After careful consideration, we feel that it has merit but does not fully meet PLOS ONE’s publication criteria as it currently stands. Therefore, we invite you to submit a revised version of the manuscript that addresses the points raised during the review process.

Major Revisions Required

We look forward to receiving your revised manuscript.

Kind regards,

Verda Salman, PhD

Academic Editor

PLOS ONE

Journal Requirements:

2. You indicated that ethical approval was not necessary for your study. We understand that the framework for ethical oversight requirements for studies of this type may differ depending on the setting and we would appreciate some further clarification regarding your research. Could you please provide further details on why your study is exempt from the need for approval and confirmation from your institutional review board or research ethics committee (e.g., in the form of a letter or email correspondence) that ethics review was not necessary for this study? Please include a copy of the correspondence as an "Other"" file.

3.Please provide additional details regarding participant consent. In the ethics statement in the Methods and online submission information, please ensure that you have specified what type you obtained (for instance, written or verbal, and if verbal, how it was documented and witnessed). If your study included minors, state whether you obtained consent from parents or guardians. If the need for consent was waived by the ethics committee, please include this information.

5.  We note that you have stated that you will provide repository information for your data at acceptance. Should your manuscript be accepted for publication, we will hold it until you provide the relevant accession numbers or DOIs necessary to access your data. If you wish to make changes to your Data Availability statement, please describe these changes in your cover letter and we will update your Data Availability statement to reflect the information you provide

6. We note that Figure 1 in your submission contain [map/satellite] images which may be copyrighted. All PLOS content is published under the Creative Commons Attribution License (CC BY 4.0), which means that the manuscript, images, and Supporting Information files will be freely available online, and any third party is permitted to access, download, copy, distribute, and use these materials in any way, even commercially, with proper attribution. For these reasons, we cannot publish previously copyrighted maps or satellite images created using proprietary data, such as Google software (Google Maps, Street View, and Earth). For more information, see our copyright guidelines: http://journals.plos.org/plosone/s/licenses-and-copyright.

Additional Editor Comments:

Major Revisions required

Reviewers' comments:

Reviewer's Responses to Questions

**Comments to the Author**

1. Is the manuscript technically sound, and do the data support the conclusions?

Reviewer #1: Yes

Reviewer #2: No

2. Has the statistical analysis been performed appropriately and rigorously? 

Reviewer #1: Yes

Reviewer #2: No

3. Have the authors made all data underlying the findings in their manuscript fully available?

Reviewer #1: Yes

Reviewer #2: No

4. Is the manuscript presented in an intelligible fashion and written in standard English?

Reviewer #1: Yes

Reviewer #2: No

5. Review Comments to the Author

Reviewer #1: Well conducted and described survey. Manuscript provides detailed statistical analysis. The recommendations provided could be a great assert for the enforcement agencies and policy makers. Further, this kind of surveys could be conducted in the other parts of country where, prevalence of tobacco consumption is high in adolescent age group.

Reviewer #2: 1. Introduction section does not highlight the gap in literature and the contribution of the study. It seems like that the data collection is the only contribution. Reader is unable to identify the objectives and research question of the study.

2. Size variable consider less than 30 students in a school- is there exists any school in India with student population of less than 30?

3. P#6, authors state, “Schools from all 133 districts and 1,763 Blocks (out of 1868) of the four States reported activity on the App.” If this true, then there is no district under consideration for the comparison group?

4. P#6, “On logistic regression modelling, primary school category had significant higher T-shops as compared to secondary (OR: 0.835) and higher secondary (0.829)”. These results are counter intuitive and counter factual as well.

5. On balance, the study concludes that girls only and boys only schools have higher chances of having T-shops near the schools. These results are interpreted mechanically without relating to theory and hence the authors are unable to provide any rationale to support the findings of the study.

6. The study relies on the reported results and does not providing any mechanism regarding the reporting biases- both type-I & II errors.

7. Results are poorly presented- screenshots of the table are given in the appendices.

6. PLOS authors have the option to publish the peer review history of their article (what does this mean?). If published, this will include your full peer review and any attached files.

Reviewer #1: No

Reviewer #2: No

---

## [Author Response · Author response to Decision Letter 0]

10 Sep 2023

Each comment has been responded to including the one received on 28th July. The 'Response to Reviewers' file has been uploaded in the previous section. 

Many thanks for the suggestions and comments. It really helps.

---

## [Decision Letter · Decision Letter 1]

31 Jan 2024

PONE-D-22-32913R1SURVEY FROM 61,748 SCHOOLS IN FOUR STATES OF INDIA ON SALE OF TOBACCO PRODUCTS NEAR SCHOOLSPLOS ONE

Dear Dr. Sarin,

Thank you for submitting your manuscript to PLOS ONE. After careful consideration, we feel that it has merit but does not fully meet PLOS ONE’s publication criteria as it currently stands. Therefore, we invite you to submit a revised version of the manuscript that addresses the points raised during the review process.

We look forward to receiving your revised manuscript.

Kind regards,

Verda Salman, PhD

Academic Editor

PLOS ONE

**Additional Editor Comments:**

Major revisions are suggested.

Please look into the comments given by one of the reviewers and address them.

1. Regression results indicate that the possibility of T-shops near primary schools is significantly higher than upper primary and secondary schools, which is against reality. It seems like either data cleaning is not done appropriately or the regression analysis is not conducted accurately.

2. Similarly, the Chi-Square results show that more than 98 percent of shops are near co-education schools, whereas the regression results are not in line with these findings.

Reviewers' comments:

Reviewer's Responses to Questions

**Comments to the Author**

1. If the authors have adequately addressed your comments raised in a previous round of review and you feel that this manuscript is now acceptable for publication, you may indicate that here to bypass the “Comments to the Author” section, enter your conflict of interest statement in the “Confidential to Editor” section, and submit your "Accept" recommendation.

Reviewer #2: (No Response)

2. Is the manuscript technically sound, and do the data support the conclusions?

Reviewer #2: No

3. Has the statistical analysis been performed appropriately and rigorously? 

Reviewer #2: No

4. Have the authors made all data underlying the findings in their manuscript fully available?

Reviewer #2: Yes

5. Is the manuscript presented in an intelligible fashion and written in standard English?

Reviewer #2: Yes

6. Review Comments to the Author

Reviewer #2: 1. Regression results indicates that the possibility of T-shops near primary schools is significantly higher than upper primary and secondary schools which is against the reality. It seems like that either data cleaning is not done appropriately or the regression analysis is not conducted accurately.

2. Similarly, the Chi-Square results shows that more than 98 perccent shops are near co-eduation shcool whereas the regression results are not inline with these findings.

7. PLOS authors have the option to publish the peer review history of their article (what does this mean?). If published, this will include your full peer review and any attached files.

Reviewer #2: No

---

## [Author Response · Author response to Decision Letter 1]

20 Mar 2024

RESPONSE TO QUERY (31st January 2024)

1. Regression results indicate that the possibility of T-shops near primary schools is significantly higher than upper primary and secondary schools, which is against reality. It seems like either data cleaning is not done appropriately or the regression analysis is not conducted accurately.

Thank you for the query. You are right, as the data suggests that in two states, the T-shops are higher in the upper primary and secondary schools as compared to primary, however, in pooled analysis, T-shops around primary schools are higher (Table 8).

2. Similarly, the Chi-Square results show that more than 98 percent of shops are near co-education schools, whereas the regression results are not in line with these findings.

According to Table 7, Chi-Square test indicates that there was a significant difference between the number of shops between co-education schools and other gender, however, the number of schools with tobacco shops in co-education schools (15790 / 60321) 26 percent while for girls (31.6 percent). Therefore, the results say the same in regression with girls and boys having higher odds ratio as compared to co-education (Table 8). Hope this resolves your query.

---

## [Editor Report · Decision Letter 2]

25 Mar 2024

SURVEY FROM 61,748 SCHOOLS IN FOUR STATES OF INDIA ON SALE OF TOBACCO PRODUCTS NEAR SCHOOLS

PONE-D-22-32913R2

Dear Dr. Sarin,

We’re pleased to inform you that your manuscript has been judged scientifically suitable for publication and will be formally accepted for publication once it meets all outstanding technical requirements.

Kind regards,

Verda Salman, PhD

Academic Editor

PLOS ONE

Additional Editor Comments (optional):

Accepted for Publication
---

## [Editor Report · Acceptance letter]

4 Apr 2024

PONE-D-22-32913R2 

PLOS ONE

Dear Dr. Sarin, 

I'm pleased to inform you that your manuscript has been deemed suitable for publication in PLOS ONE. Congratulations! Your manuscript is now being handed over to our production team.

Kind regards, 

on behalf of

Dr. Verda Salman 

Academic Editor

PLOS ONE